# Adding neuroplasticity to a CNN-based in-silico model of neurodegeneration

**Jasmine A. Moore**
Department of Biomedical Engineering
University of Calgary
Calgary, AB
jasmine.moore@ucalgary.ca

**Matthias Wilms**
Department of Radiology
University of Calgary
Calgary, AB
matthias.wilms@ucalgary.ca

**Kayson Fakhar**
Institute of Computational Neuroscience
University Medical Center UKE
Hamburg, Germany
k.fakhar@uke.de

**Fatemeh Hadaeghi**
Institute of Computational Neuroscience
University Medical Center UKE
Hamburg, Germany
f.hadaeghi@uke.de

**Claus C. Hilgetag**
Institute of Computational Neuroscience
University Medical Center UKE
Hamburg, Germany
c.hilgetag@uke.de

**Nils D. Forkert**
Department of Radiology
University of Calgary
Calgary
nils.forkert@ucalgary.ca

## Abstract

The aim of this work was to enhance the biological feasibility of a deep convolutional neural network-based *in-silico* model of neurodegeneration of the visual system by adding neuroplasticity to it. Therefore, deep convolutional networks were trained for object recognition tasks and progressively lesioned to simulate the onset of posterior cortical atrophy, a condition that affects the visual cortex in patients with Alzheimer's disease (AD). After each iteration of injury, the networks were retrained on the training set to simulate the continual plasticity of the human brain, when affected by a neurodegenerative disease. More specifically, the injured parts of the network remained injured while we investigated how the added retraining steps were able to recover some of the model's baseline performance. The results showed that with retraining, a model's object recognition abilities are subject to a smoother decline with increasing injury levels than without retraining and, therefore, more similar to the longitudinal cognition impairments of patients diagnosed with AD. Moreover, with retraining, the injured model exhibits internal activation patterns similar to those of the healthy baseline model compared to the injured model without retraining. In conclusion, adding retraining to the *in-silico* setup improves the biological feasibility considerably and could prove valuable to test different rehabilitation approaches *in-silico*.

## 1 Introduction

Deep learning models have not only emerged as integral tools for solving many complex classification, regression, and object recognition problems, but are also increasingly explored as potential tools to model information processing in the brain (Lo Vercio et al., 2020; Yamins & DiCarlo, 2016). Deep

4th Workshop on Shared Visual Representations in Human and Machine Visual Intelligence (SVRHM) at the Neural Information Processing Systems (NeurIPS) conference 2022. New Orleans.

convolutional neural networks (DCNNs), a deep learning model architecture specifically designed for solving computer vision problems, were originally inspired by the neuron and synaptic structure found in the mammalian visual cortex (Rawat & Wang, 2017). The concepts used to inspire DCNNs date back to early models of the visual system, postulated by Hubel and Wiesel (Hubel & Wiesel, 1962, 1968). A recent developing avenue that is still in its infancy is the use of deep learning models as an abstraction of a cognitively healthy human brain, which is used as the basis for simulating neurodegenerative diseases (Tuladhar et al., 2021; Moore et al., 2022). In this work, we build upon previously established work using CNNs for simulating posterior cortical atrophy (PCA), a condition that can accompany Alzheimer's disease (AD). PCA is caused by the accelerated degeneration and thinning of visual cortical areas (i.e., V1, V2, V3, V4) and results in patients experiencing a loss of visual recognition abilities (da Silva et al., 2017; Crutch et al., 2012). Since DCNNs were specifically designed for object recognition and modelled following information processing in the mammalian brain, neuronal injuries as seen in PCA can be intuitively modelled in DCNNs. DCNNs and human neural activation data have been compared in various works, where the popular VGG19 model has been found to exhibit one of the highest similarity metrics in terms of the Brain Score and Brain Hierarchy Score (Schrimpf et al., 2018; Nonaka et al., 2021). Thus, in this study, we use VGG19 as a baseline, cognitively healthy object recognition model.

While model compression and pruning is an active branch of deep learning research, our paradigm does not follow typical pruning methods that aim to reduce the number of parameters in a model while retaining full function (Choudhary et al., 2020). In contrast, we use progressive unstructured random pruning and retraining to simulate the cognitive effects of a neurodegenerative disease as a function of abnormal levels of atrophy. Previous research has established parallels between synaptic and neuronal pruning and the onset of posterior cortical atrophy (Moore et al., 2022). This study expands upon and improves Moore et al. (2022) by adding the crucial mechanism of simulated neuroplasticity as seen in Figure 1. We find that by adding iterative retraining with every pruning step of synaptic ablation, the decline of visual cognition is much smoother and more similar to what is seen in patients with AD (Mattsson et al., 2017).

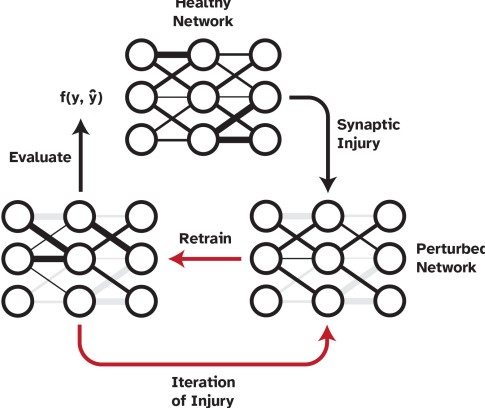

Figure 1: Pipeline of progressive synaptic injury with the added mechanism of neuroplasticity. After each iteration of synaptic damage, the model is retrained on the original training split of data and evaluated.

## 2 Methods and Materials

### 2.1 Models and Data

The basis for the cognitively healthy object recognition model is a VGG19 model trained on the CIFAR10 dataset (Russakovsky et al., 2015). The VGG19 architecture is comprised of five convolutional blocks, each followed by a max-pooling layer, and three fully-connected dense layers ending in a softmax activation with ten nodes corresponding to the ten respective classes in the dataset. CIFAR10 is a commonly used dataset for computer vision research made up of 60,000 32x32 natural

color images. The train/test split is 50,000 and 10,000 images, respectively. The dataset consists of ten classes: plane, car, bird, cat, deer, dog, frog, horse, ship, and truck, with 6000 images in each class. Our model was trained for 10 epochs with a learning rate of 0.001, using a batch size of 40, and a stochastic gradient descent optimizer. After training the model performs with a mean class-wise accuracy of 92.3% on the test set of images.

## 2.2 Synaptic Ablation and Retraining

Synaptic ablation was imposed on the network in a uniformly disperse and progressive manner as proposed in Moore et al. (2022). This 'injury' was implemented by setting weights from convolutional layers and dense layers in the network to zero, effectively severing the connections between nodes. This is akin to progression of synaptic damage seen in neurological diseases that accelerate atrophy rates in the brain such as posterior cortical atrophy. We imposed synaptic ablation at a step rate of $(1 - (1 - \gamma))^n$ where $\gamma$ is the relative fraction of weights being ablated to the remaining uninjured weights in the network, and $n$ is the number of iterations of injury. In our experiments, $\gamma$ is set to 0.2 (20% of weights ablated) and n is set to 15 iterations. Following each iteration of injury, we retrained the model on the training split of data using the same initial training parameters for 3 epochs to investigate how model performance could be regained.

## 2.3 Representational Dissimilarity Matrices

Representational dissimilarity matrices (RDMs) were computed to examine the changes in internal activations and representations of categorized data of both the injured and retrained networks when compared to the healthy network. RDMs were generated by pairwise comparison between activations of the network's penultimate layer for all test set images using Pearson's correlation coefficient. We constructed RDMs for each iteration of both network ablation and retraining, and then compared them to the healthy network's RDM using Kendall's tau correlation coefficient. This effectively enabled us to quantify the effects of both injury and retraining on internal activations of the network.

# 3 Results

## 3.1 Accuracy

The results showed that model performance is immediately affected by the first step of random synaptic injury (20% of synapses), leading to a large drop in mean class-wise accuracy from 92.3% accuracy achieved by the healthy model on the test set to a mere 14.3%. Interestingly, accuracy was massively regained after 3 epochs of training and was restored to 91.9% after the first training session. This pattern continued to repeat with each iteration of injury leading to a massive drop in accuracy, with the model tending to perform only at chance level (10% accuracy). However, retraining continued to improve model accuracy by a large margin, until 91% of initial synapses had been removed. After this point, the model could not compensate for the continuing imposed damage. These effects are shown in Figure 2A.

## 3.2 Representational Dissimilarity Matrices

In line with the results of the model accuracy evaluation, the internal representations of the model were able to regenerate and recover with retraining after injury. In the first iteration of damage and retraining (20% of synapses ablated), the injured model's correlation to the healthy RDM revealed a Kendall's tau of 0.18. After retraining for three epochs, the model was able to reconstruct activations more similar to those of the healthy model, resulting in a Kendall's tau value of 0.71. Figure 2B shows how retraining after each injury step leads to regaining category-distinguishable activations and a smooth cognitive decline. A qualitative examination also reveals how the network activations are affected through injury and retraining. As seen in Figure 3, the uninjured network initially had clearly defined activations grouped according to classes in the CIFAR10 dataset. Upon injury, the network lost this categorical representation and the RDMs became noisy. After retraining, however, categorical structure between the classes is regained. This trend continues progressively as injury and retraining steps are applied up until 89.3% of the network synapses are ablated, at which point there no longer remains a difference in Kendall's tau correlation between injured and retrained RDMs.

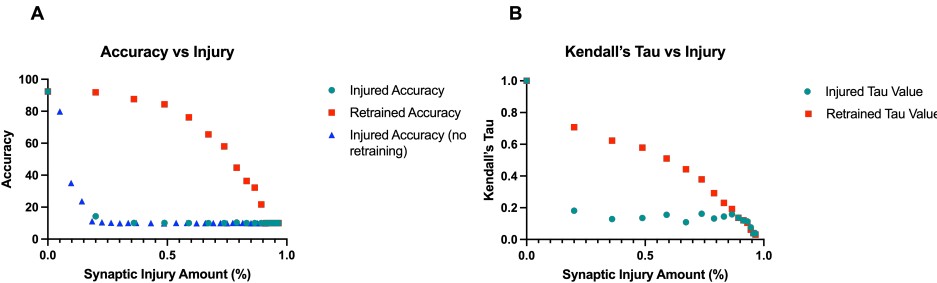

Figure 2: (A) Model accuracy as a function of progressive damage and retraining. The model immediately has a substantial drop in accuracy after 20% of synapses are removed but regains function after retraining on the training set. (B) Kendall's tau is plotted as a function of injury to compare injured and retrained RDMs to the healthy network's RDM. A perfect correlation is Kendall's tau = 1. As injury is imposed, the RDMs become much less correlated; however, this graph demonstrates how retraining allows the model to recover internal activation patterns more like the healthy model.

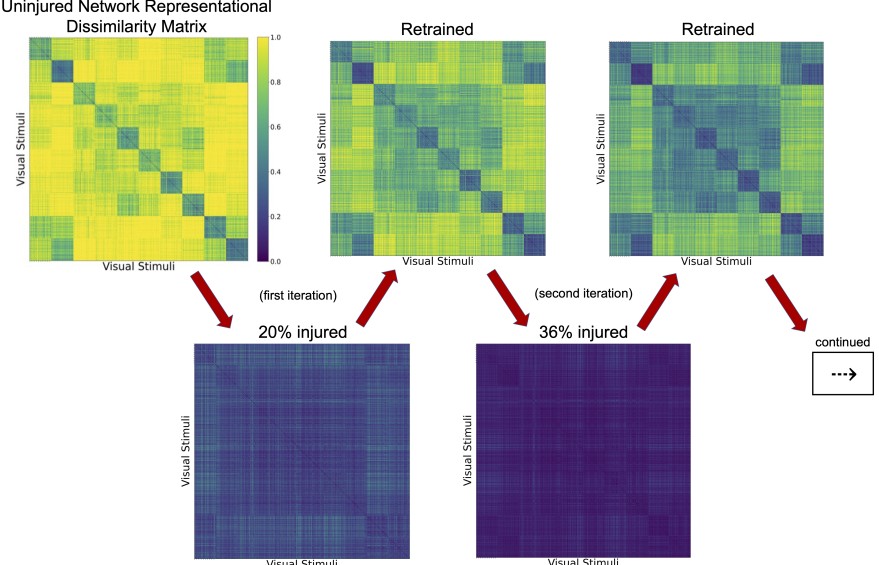

Figure 3: A qualitative examination of RDMs as two iterations of injury and retraining are completed. Before injury, the model has distinct activations for each of the ten different classes. As injury and retraining is imposed, the activations first lose their categorical nature but are able to recover some of it with retraining.

## 4  Discussion

The main finding of this study is that with the incorporation of retraining to simulate neuroplasticity after the progression of injury, the model's object recognition abilities progressively decline at a much smoother and slower rate than without retraining. This slow decline is akin to the degradation of cognitive abilities seen in patients with PCA and AD (Fox et al., 1999; Hodges et al., 1995; Jefferson et al., 2006). Previous works using DCNNs to model neurodegenerative diseases used a static injury paradigm that led to extreme loss of object recognition abilities even with low levels (i.e., 15-20%) of synapses injured (Tuladhar et al., 2021; Moore et al., 2022). Expanding upon this work, here we developed a framework that is able to simulate irreversible injury, while the unaffected filters and weights were subjected to 're-learning' processes using training data. We found that the retrained

model is able to compensate for the damaged pathways (synapses) and reconstructs the original activation patterns of the healthy model when presented with images in the test set. We believe that the introduction of the biologically important concept of neuroplasticity, which equips our DCNN with a retraining mechanism, can be seen as an important step towards developing biologically more meaningful in silico models of PCA. The development of this field of work has the potential to lead to positive societal implications by increasing understanding around the progression of these diseases. There are currently no foreseen negative impacts.

One limitation of this work is the rather simplistic dataset that was used for the experiments. CIFAR10 only contains ten relatively easily distinguishable classes of images. To capture the true capacity of human vision more accurately, a dataset with considerably more complexity will be used in future work. Another limitation are some notable differences in information processing between DCNNs and the biological visual system (Lonnqvist et al., 2021). For example, in DCNNs a single convolutional kernel is used to cover the whole visual receptive field. However, while this remains to be true, the object recognition performance of DCNNs is comparable to that of humans and DCNNs have the ability to predict neural activation in the primate visual cortex better than any other method to date (Cadieu et al., 2014; Yamins et al., 2014).

Crucial future directions for this work will be to further investigate the details surrounding the iterative retraining process, as well as more realistically represent disease progression. This investigation will allow for the exploration of rehabilitation strategies in terms of what methods of retraining enable in silico models to regain the most function. For example, we can provide models with training data that is directly related to the types of errors the models begin to make with initial injury. This could be compared against re-training strategies that would simply re-use all initial training data. In addition, it may be important to evaluate the effects of other variables such as training the network on new data rather than previously seen data, or adjusting the number of epochs used in one iteration of retraining. By probing these types of differences in network plasticity and recovery, it may be possible to identify optimal intervention strategies and relate these findings to rehabilitation techniques used in patients with dementia.

## Acknowledgments and Disclosure of Funding

This work was supported by the Canada Research Chairs program, the River Fund at Calgary Foundation, Natural Sciences and Engineering Research Council of Canada (NSERC), Alberta Innovates – Data Enabled Technologies, and from the NSERC – Hotchkiss Brain Institute Brain CREATE program. The funding agencies had no role in the study design, collection, analysis and interpretation of data, nor preparation, review or approval of the manuscript.

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
