# OpenReview forum: "Adding neuroplasticity to a CNN-based in-silico model of neurodegeneration"
_NeurIPS.cc/2022/Workshop/SVRHM — SVRHM Poster_

### Official Review · Reviewer_bm3Q · 2022-10-14
**Review of Adding neuroplasticity to a CNN-based in-silico model of neurodegeneration.**

**Rating:** 6
**Confidence:** 4

**Review:**

## Summary:
In the present paper, the authors expand previous work (Moore et al., 2022) and incorporate a retraining step after each synaptic injury in a DCNN to simulate the neuroplasticity behavior of the brain, when affected by a neurodegenerative disease. Their experiments used a VGG19 network fine-tuned on the CIFAR10 dataset and showed that the “retraining” step generates a smoother decline with increasing injury levels.

## Strengths:
+ The incorporation of the “neuroplasticity” in DCNN looks like an interesting research direction.
+ The use of DCNN in clinical application is necessary and valuable.

## Weaknesses:
+ There is a lack of error reporting. As previous work has stated, there should be multiple runs and they should be initialized with a different set of weights in order to reduce potential biases.
+ Although I understand that this is a work in progress, I would have appreciated seeing the performance of other CNNs to evaluate if the effect of the “retraining” step helps by itself or if there is a dependence on the size|type of architecture.
+ As mentioned in the limitations of the paper,  it would be interesting to test this approach in other datasets to evaluate the impact of the size in the analysis.

## Suggestion to Authors
+ I would suggest looking into other architectures that have been shown to resemble better visual processing in primates at brain score. You can take a look at different models from its online platform and take for example a hybrid ANN with a V1 front-end computation (https://www.biorxiv.org/content/10.1101/2020.06.16.154542v2)
+ Previous work (Dapello et al.,2020, Berrios & Deza 2022) has shown that adversarial training helps to increase the brain-score of architectures that are not biologically plausible (e.g Transformers). I would suggest testing these robust architectures.
+ I encourage authors to look into the Center Kernel Alignment (https://arxiv.org/abs/1905.00414) score to compare the internal representation of deep neural networks.

Overall, I think this paper requires more experiments with different datasets and architectures to clearly assess the real impact of the “retraining” step. However, I think the paper could generate interesting discussions relevant to the workshop's audience and the authors.

---

### Official Review · Reviewer_C46k · 2022-10-14
**Adding learning to in-silico models of neurodegeneration**

**Rating:** 6
**Confidence:** 3

**Review:**

In this paper, the authors used deep neural networks as in-silico models for understanding posterior cortical atrophy. Building on prior work that damaged (lesioned) neural connections to mimic the effects of brain damage, the present study explored whether retraining can allow a damaged visual system to regain both function and representational similarity to healthy (non-damaged) brains. The authors observed that when damaged networks are allowed to continue learning, the model’s performance shows a smooth decline with increasing amounts of brain damage, akin to longitudinal cognition impairments seen in patients diagnosed with Alzheimer’s disease.

I liked this paper. The general goal of extending the reverse engineering approach to neurodegeneration seems promising and will likely be important for the future expansion of the broader scientific paradigm. This paper adds a novel element to the literature by allowing models to continue learning after brain damage, thereby closing the gap between the learning capacities of humans and machines.

Comments:
1. This paper would be strengthened by the addition of more models, exploring whether these effects generalize across a range of different architectures and legion types (e.g., local vs. distributed damage). As the authors note, it would also be nice to see how these results generalize to different training data and tasks. As such, the present paper feels like a promising starting point towards a more comprehensive paper that explores how adding plasticity to deep neural networks can improve in-silico models of neurodegeneration.

---

### Official Review · Reviewer_pxH8 · 2022-10-14
**Interesting results with potential applications for modeling neurodegeneration**

**Rating:** 7
**Confidence:** 4

**Review:**

Summary: This work builds on previous research which modeled posterior cortical atrophy (PCA) in CNNs by investigating the role neuroplasticity plays in the progressive decline in object recognition performance. The results clearly show how retraining after a portion of the weights in the network are ablated leads to a smoother decline in object recognition performance compared to networks that are not retrained after injury. Additionally, the RDMs of the retrained networks are more similar to the uninjured model compared to the injured models without retraining suggesting that adding neuroplasticity allows for some retention of the class representations in the network.

Strong Points:
1. The analysis of the RDMs using Kendall’s Tau correlation provides strong evidence that retraining helps retain distinct class representations in the network.
2. The biological inspirations behind VGG19 make it a good model to study neurodegeneration since the types of computations are more similar to the biological system in primary visual cortex (V1).

Suggestions:
1. Run more trials (maybe 10 or so) to improve the statistical certainty of the results.
2. Ensure the model is properly converged. The use of a learning rate scheduler allows for a more progressive decline in the loss landscape during training. The accuracy reported in the study (92.3%) is a bit below the SOTA for VGG19 (93.7% Sterneck et al. 2021). Along these same lines, it might be interesting to see how the train/test loss changes with retraining. These data would also clarify if the decrease in accuracy after retraining is solely because of the loss of synapses or if perhaps the model is under-converged after retraining.
3. Try more complex architectures which have a higher Brain Score. For example, recent studies have shown transformers (Berrios and Deza 2022) and Efficient-Net (Riedel 2022) have some of the highest Brain Scores.
4. Compare the activations in the network to neural data. This type of analysis would clarify how similar this network really is to biological neural networks damaged by AD and PCA.
5. Clarify the rehabilitation strategies. I was a bit confused by the suggestions given at the end of the paper regarding retraining the networks on data that the injured model tends to make errors on since the results presented suggest that the decrease in accuracy is uniform across all classes.

---

### Official Review · Reviewer_28U9 · 2022-10-15
**Well-written. Obvious results.**

**Rating:** 6
**Confidence:** 4

**Review:**

This paper uses a CNN-based in-silico model of neurodegeneration and is able to demonstrate a smoother decline with increasing levels of injury using the principle of neuroplasticity. In more detail, the authors first train a CNN on an object recognition task and then progressively lesion the network at random. After each iteration of injury, the model is allowed to retrain on the same training dataset. This retraining helps the model regain a significant fraction of its performance after injury.

This paper is well-written and the study is well-motivated. The results back its claim. However, this study suffers from a major weakness. The result of this study is obvious since the CNN used is overly parametrized. It is safe to assume that the network used isn't highly regularized since the authors do not mention the use of any regularization such as weight decay or dropout. Therefore, It is not a surprise that such a network is able to recover most of its performance through retraining after injury. The study could benefit from using a smaller or highly regularized CNN.